# Do Victims Really Help Their Abusive Supervisors? Reevaluating the Positive Consequences of Abusive Supervision

**DOI:** 10.3390/bs13100815

**Published:** 2023-10-03

**Authors:** Wen Pan, Li-Yun Sun

**Affiliations:** 1Faculty of Hospitality and Tourism Management, Macau University of Science and Technology, Macao 999078, China; wpan@must.edu.mo; 2School of Business, Macau University of Science and Technology, Macao 999078, China

**Keywords:** abusive supervision, leader–member exchange, workplace self-blame, workplace guilt, supervisor-directed helping

## Abstract

Do victims really help their abusive supervisors? Does abusive supervision have any positive consequence? The study aims to address this concern through extending the work by Tröster and Van Quaquebeke (2021). Using subordinates’ self-reports, Tröster and Van Quaquebeke (2021) found that abusive supervision in high-quality leader–member exchange (LMX) relationship motivates subordinates to blame themselves, subsequently making them feel guilty and make up for it by being more helpful. By integrating both subordinates’ and supervisors’ perspectives, and using multi-wave, multi-source, and multi-level data collected in China, we obtain three major findings. First, as a replication of their findings, LMX moderates the direct effect of abusive supervision on workplace self-blame, and the indirect effect of abusive supervision on workplace guilt via workplace self-blame. The positive direct and indirect effects are stronger when LMX quality is higher. Second, different from their findings, LMX moderates the indirect effect of abusive supervision on supervisor-directed helping (evaluated by supervisors) via workplace self-blame and workplace guilt such that the negative indirect effect is stronger when LMX quality is higher. Third, as an extension, supervisor-evaluated LMX (SLMX) moderates the effect of workplace guilt on supervisor-directed helping such that the negative effect is stronger when SLMX is lower-quality. Put together, LMX and SLMX moderate the indirect effect of abusive supervision on supervisor-directed helping via workplace self-blame and workplace guilt. The negative indirect effect is stronger when LMX quality is higher, but SLMX quality is lower. Our study challenges previous speculations on the positive or beneficial consequences of abusive supervision, and thus contributes to the literature on abusive supervision.

## 1. Introduction

How does abusive supervision influence employees and supervisors? Research suggests the negative effect of abusive supervision on desirable outcomes or a positive effect on undesirable outcomes (see meta-analysis or review by [1,2,3]). As such, “the preponderance of work to date suggest[s] that abusive supervision undermines individual, unit, and organizational functioning” ([4], p. 145). Abusive supervision is defined as “subordinates’ perceptions of the extent to which supervisors engage in the sustained display of hostile verbal and nonverbal behaviors, excluding physical contact” [5] (p. 178). In this stream of research, there appears some studies treating abusive supervision as acceptable because abusive supervision may improve employee performance in some contexts. For example, some scholars argued that a moderate level of abusive supervision can stimulate employee creativity [6], and some maintained that daily abusive supervisor behavior interacts with subordinates’ attribution tendencies of leaders’ performance promotion motives to predict subordinates’ task performance [7]. However, the argument could be, to some extent, conceived as a justification for unethical leadership behavior [8]. In these studies, abusive leaders obviously care more about economic or instrumental outcomes [7] and less about subordinates’ self-esteem and growth in the organization. The simple and rude behavior cannot effectively improve subordinates’ task and creative performance.

Recently, there have been some other voices claiming that abusive supervision can have beneficial consequences for leaders (e.g., Tröster and Van Quaquebeke (2021) [9]. Following the well-established view on the negative impact of abusive supervision, our study was motivated to dwell on the positive consequence of abusive supervision for leaders. Specifically, we take dual perspectives in examining the consequence of abusive supervision. Tröster and Van Quaquebeke (2021) [9] relied on subordinates’ self-reports of supervisor-directed helping and relationship quality with their supervisors. We attempt to dig deeper into their study through involving supervisors in assessing their subordinates’ helping behavior and their relationship quality with subordinates. We suggest that the finding of positive consequences of abusive supervision has serious limitations. Our work is important because it gives a balanced view by integrating both subordinates’ and supervisors’ perspectives in investigating the effect of abusive supervision. 

The study contributes to the literature on abusive supervision by re-evaluating the positive consequences of abusive supervision, thereby advancing a nuanced understanding of how abusive supervision influences subordinates and supervisors themselves. We extend Tröster and Van Quaquebeke’s (2021) work by adding a supervisor perspective in the supervisor-subordinate interactions. We argue that supervisors perceive those abused subordinates as less helpful, which challenges previous speculations that “leaders may engage in abusive supervision because it has beneficial consequences for them” [9] (p. 1793).

## 2. Literature Review and Hypotheses

Tröster and Van Quaquebeke’s (2021) [9] study is the most significant piece of research that claims that abusive supervision can have beneficial consequences for leaders. We address the issue by reevaluating their study and retesting their hypotheses. Tröster and Van Quaquebeke (2021) [9] explored the mechanism through which abusive supervision influences supervisor-directed helping from a subordinate perspective. Specifically, “abusive supervision in high LMX dyads motivates people to preserve these kinds of relationships by blaming themselves for having done something wrong that might have jeopardized this relationship, subsequently feeling guilty, and making up for it by being more helpful” [9] (p. 1808). Leader–member exchange (LMX) indicates the general relationship quality between subordinates and their supervisors. Tröster and Van Quaquebeke’s (2021) [9] theorizing is reasonable because subordinates attribute their supervisors’ maltreatment to something caused by their own fault, particularly when they think that they have a high-quality LMX with their supervisors. A high-quality LMX suggests that subordinates receive more supervisory support and guidance, such as supervisors’ liking, respect, and trust [10,11]. (Gerstner & Day, 1997; Graen & Uhl-Bien, 1995). In associating workplace guilt with supervisor-directed helping, Tröster and Van Quaquebeke (2021) used a social functional approach of guilt [9]. According to their approach, guilt patterns appear to be strongest and most common in the context of communal and dyadic relationships, which are characterized by expectations of mutual concern between supervisors and subordinates. Guilt can serve relationship-enhancing functions. From the subordinates’ perspective, subordinates feel guilty for failing to live up to their supervisors’ expectations and may suffer from emotional distress. Therefore, “they will alter their behavior (to avoid guilt) in ways that seem likely to maintain and strengthen the relationship” [12] (p. 247). They help their supervisors to repair their relationship after being abused.

We concur with Tröster and Van Quaquebeke’s (2021) argumentation that “abusive supervision in a high LMX dyad increases the likelihood that employees look within themselves for the cause of the abuse, blame themselves, and then feel guilty.” (p. 1796). We thus repeat their following two hypotheses.

**Hypothesis 1.** 
*The positive effect of abusive supervision on workplace self-blame is stronger when LMX quality is high (vs. low).*


**Hypothesis 2.** 
*The positive relationship between abusive supervision and workplace guilt is mediated by workplace self-blame. This indirect relationship is stronger when LMX quality is high (vs. low).*


For all that, however, we need to caution that the effect of abusive supervision on workplace guilt is temporal, and Tröster and Van Quaquebeke’s (2021) [9] argumentation represents a subordinate perspective. We challenge the generalizability of their findings for the following three reasons. First, as supervisors’ frequent, sustained, hostile behaviors inevitably damage good relationships with their subordinates, workplace self-blame will not be successfully generated by repeated abusive supervision. Further, workplace guilt created by workplace self-blame will gradually be weakened. Thus, the expected supervisor-directed helping caused by workplace guilt will not happen as expected after repetitive hostile supervisor behaviors. As a support for the argument, scholars contended that the functional effects of abusive supervision do not endure over time [13] and abusive behaviors can be costly [3]. Second, in a communal, dyadic relationship, supervisors are supposed to treat their subordinates well and repair damage to a relationship arising from a transgression [12]. In this situation, their abusive supervision and subordinates’ guilt may increase the probability of redistributing emotional distress [12], and supervisors may suffer from emotional distress due to their abusive supervision. The emotional distress on the subordinates’ part may motivate supervisors to treat subordinates well, avoid transgressions in the future, and enable less powerful subordinates to get their way [12]. Abusive supervision cannot be justified or sustained in the high-quality LMX relationship. Third, social exchange theory [14] suggests that people are obliged to return an equivalent behavior for something given [15]. “Abusive supervision is likely to trigger a poor exchange between supervisors and subordinates [14], whereby abused subordinates may reciprocate their supervisors by withholding their efforts at work.” [16] (p. 532). Meta-analysis and review studies suggest that abusive supervision is negatively related to supervisors’ ratings of their subordinates’ in-role and extra-role performance. Indeed, when subordinates report a high level of abusive supervision, these subordinates are evaluated by their supervisors to have low levels of helping directed at their organizations and coworkers [16]. Consistently, supervisors evaluate subordinates who are abused as less helpful to them. Workplace guilt, as a mediating emotional state, does not reverse the negative relationship between subordinate-reported abusive supervision and supervisors’ evaluation. Although subordinates feel guilty for their work, supervisors do not change their evaluation of these subordinates whatsoever. The contrasting effect between subordinates’ response and supervisors’ independence in the evaluation is enhanced when subordinates have a stronger sense of guilt triggered by abusive supervision in their perceived high-quality LMX relationship. Following this reasoning, we differ from Tröster and Van Quaquebeke’s (2021) [9] study by proposing following hypothesis.

**Hypothesis 3.** 
*The relationship between abusive supervision and supervisor-directed helping (rated by supervisors) is serially mediated by workplace self-blame and workplace guilt. The negative indirect effect of abusive supervision is stronger when LMX quality is high (vs. low).*


We have argued that subordinates’ guilt may be negatively related to supervisors’ rating of their subordinates’ helping behavior. This negative relationship can be more evident when supervisors have a low-quality relationship with them. Supervisor-reported LMX (SLMX) refers to supervisors’ evaluation of overall relationship quality with their subordinates. In high-quality SLMX, supervisors maintain general exchange relationships with their subordinates, characterized by mutual trust, respect, and support. In low-quality SLMX, there is a lack of trust, respect, and support among supervisors and their subordinates. Our study suggests that the supposed workplace guilt mechanism linked to supervisor-directed helping does not function in the low-quality SLMX context. First, with low-quality SLMX, supervisors do not expect the workplace guilt influence technique to be successful and, therefore, do not often operate it in the workplace to win help from their subordinates. Second, when supervisors have lower-quality relationship with their subordinates, they rate those abused subordinates even worse and as less helpful. This is because in a low-quality SLMX context, subordinates do not have good communication with supervisors regarding their jobs, fail to understand supervisors’ needs and problems, and do not ask supervisors to help out in busy situations [17]. Third, workplace guilt is less likely to be produced without mutual concern, respect, and return in the relationship. When supervisors maintain a low-quality relationship with their subordinates, these subordinates tend to believe that their supervisors treat them wrongly rather than blame themselves. They thus have less motivation to restore the low-quality relationships. Their guilt-based helping behaviors will be reduced and replaced sooner or later by reciprocal behaviors, that is, negative returns to their abusive supervisor.

**Hypothesis 4.** 
*SLMX moderates the relationship between workplace guilt and supervisor-directed helping such that their negative relationship is stronger when SLMX quality is low (vs. high).*


From the subordinate perspective, the level of LMX relationship influences the extent to which subordinates blame themselves and feel guilty after being abused. The effect of abusive supervision on workplace guilt (via workplace self-blame) is stronger when LMX quality is higher. However, from the supervisor perspective, those subordinates who report being abused are rated as less helpful by their supervisors, and the indirect effect of abusive supervision on supervisor-directed helping is more negative under low-quality SLMX. The contrasting effect of abusive supervision reaches its highest when subordinates perceive a high-quality LMX relationship, while supervisors maintain a low-quality SLMX relationship with their subordinates. Integrating the moderating role of LMX and SLMX in the indirect effect of abusive supervision on supervisor-directed helping, we have the following hypothesis.

**Hypothesis 5.** 
*The indirect effect of abusive supervision on supervisor-directed helping via workplace self-blame and workplace guilt is moderated by LMX and SLMX. The negative indirect effect is stronger when LMX quality is high (vs. low) but SLMX quality is low (vs. high).*


The research model of the study is depicted in Figure 1. 

## 3. Method

### 3.1. Sample and Procedures

Research teams and survey coordinators helped collect data in three municipalities and 16 provinces in China from late January to May 2021. Survey coordinators assisted in the administration of the surveys in the participating organizations. They prepared and wrote a list of supervisors and their six immediate subordinates, and distributed questionnaires in five time points, with at least three-week time interval in between. They got back completed questionnaires one week after each wave. At Time 1, supervisors reported leader–member exchange (SLMX) and their own demo information. At Time 2, subordinates of each supervisor reported abusive supervision, leader–member exchange, and their demo information. These subordinates provided data on workplace self-blame at Time 3 and their workplace guilt at Time 4. Finally, supervisors rated supervisor-directed helping of their six immediate subordinates at Time 5.

In total, 1262 subordinate and 253 supervisor questionnaires were distributed, and 1002 subordinate and 189 supervisor questionnaires were obtained as the final sample. This results in a 79% and a 75% response rate from subordinates and supervisors, respectively. The mean age of these 1002 subordinates was 32.81 years (SD = 7.32), 49% of them were male, and they had an average educational attainment of 15.35 years (SD = 1.69). Over half of them (61.5%) were married and had average organizational tenure of 7.48 years (SD = 6.28). Of the 189 supervisors, 58% were female, had an average age of 39.58 years (SD = 7.71), and an average educational attainment of 15.84 years (SD = 1.67). The majority of them (81%) were married and had an average organizational tenure of 12.99 years (SD = 7.88).

### 3.2. Measures

Unless otherwise indicated, we used 5-point Likert scale to measure variables in the study (1 = strongly disagree, 5 = strongly agree). Alpha reliability of each variable was in Table 1.

Supervisor-evaluated LMX (SLMX, T1, reported by supervisors). We adapted 10-item scale developed by [17] to measure how supervisors evaluated their relationship quality with subordinates. We used “subordinates” to replace “other team members” or “other members in your team” in their version. One sample item was “I often make suggestions about better work methods to my subordinates”.

Abusive supervision (T2, reported by subordinate). We used the 5-item scale adapted by [18] that was originally developed by [5]. One sample item was “My supervisor ridicules me”.

Leader–member exchange (T2, reported by subordinates). A 7-item scale of leader–member exchange relationship was adapted from LMX-7 [19]. We used affirmative sentences to replace original interrogative sentences. One sample item was “My immediate supervisor understands my problems and needs”.

Workplace self-blame (T3, reported by subordinates). Consistent with [9], we define workplace self-blame as the degree to which employees blamed themselves for having done something that could have jeopardized their relationship with their supervisor in the workplace. We used the 3-item scale developed by [9] to measure workplace self-blame. One sample was “I think that I did something that jeopardized my relationship with my supervisor”.

Workplace guilt (T4, reported by subordinates). Consistent with [9], workplace guilt is defined as the degree to which employees have self-conscious emotion centered on condemning a specific workplace behavior and assuming responsibility for it. Based on prior work [20], we used a 5-item scale to measure workplace guilt. The five items were: “At work, I (1) feel regrets, (2) feel like apologizing, (3) feel ashamed, (4) feel inadequate, and (5) feel that tasks are not done well, from time to time”.

Supervisor-directed helping (T5, rated by supervisors). Following other scholars [21,22], we adapted [23] OCBI scale to measure supervisor-directed helping (or citizenship behavior). One sample was “This subordinate accepts added responsibility to help me”.

The complete items of these variables are offered in the Appendix A.

### 3.3. Data Analysis Technique

We tested our hypotheses in two steps, using Mplus 8.0 [24]. First, we tested simultaneously the individual-level moderated mediation model: LMX moderates the indirect effect of abusive supervision on supervisor-directed helping. Second, we simultaneously tested the multi-level moderated mediation model: LMX and SLMX moderate the indirect effect of abusive supervision on supervisor-directed helping. Abusive supervision, LMX, workplace self-blame, and workplace guilt were specified at level 1 (employee level), and SLMX was specified at level 2 (team level). We grand-mean centered the team-level variable SLMX. For the employee-level variables, we group-mean centered abusive supervision and LMX and used their group-mean scores to form the level-1 interaction. Because Mplus is unable to generate resampling-based bootstrapping results for a multilevel model [25], a Monte Carlo simulation approach has been used to generate a sampling distribution of the indirect effects [26]. Specifically, we constructed 95% Monte Carlo confidence interval, with 20,000 simulated parameter sets, for the indirect effects of high- and low-quality LMX and SLMX.

## 4. Results

### 4.1. Descriptive Statistics and Corrections

Result of descriptive statistics and corrections can be found in Table 1.

### 4.2. Confirmatory Factor Analysis

We conducted confirmatory factor analysis (CFA) to demonstrate the distinctiveness of six variables in the model: SLMX, abusive supervision, LMX, workplace self-blame, workplace guilt, and supervisor-directed helping. We used the parceling approach to reduce the number of SLMXs three indicators following procedures suggested or used by previous researchers [27,28].

The fit indices indicated that our hypothesized six-factor model fit the data (χ^2^ = 1321.52, *df* = 335, RMSEA = 0.054, CFI = 0.94, TLI = 0.93, SRMR = 0.046) better than alternative five-factor models. Chi-square differences of the alternative models with the hypothesized model were all significant. The results are presented in Table 2.

### 4.3. Test of Individual-Level Moderated Mediation Model

Hypothesis 1 predicted that LMX moderated the effect of abusive supervision on workplace self-blame such that the relationship would be stronger when LMX quality was high than when LMX quality was low. Table 3 (Model 1) showed a significant interaction effect between abusive supervision and LMX on workplace self-blame (b = 0.12, SE = 0.05, *p* < 0.05). Table 4 (the upper part) showed the result of a simple slope test. The simple slope test revealed that abusive supervision was more positively related to workplace self-blame when LMX quality was high (b = 0.57, SE = 0.04, *p* < 0.001) than when LMX quality was low (b = 0.39, SE = 0.06, *p* < 0.001). Their difference was significant (Δb = 0.18, SE = 0.07, *p* < 0.05). Thus, Hypothesis 1 was supported. 

Using the Johnson–Neyman technique, we displayed the conditional effect of abusive supervision on workplace self-blame at the full range of LMX from low-quality to high-quality in Figure 2. The vertical line (the value of LMX = −2.05), as shown, indicated the boundary between the significant and non-significant regions for the conditional effect of abusive supervision on workplace self-blame. The effect of abusive supervision on workplace self-blame was significant when the value of LMX was above −2.05.

Hypothesis 2 predicted that LMX moderated the indirect effect of abusive supervision on workplace guilt via workplace self-blame, such that the indirect effect would be more positive when LMX quality was high (vs. low). Table 4 showed that abusive supervision was more positively related to workplace guilt via workplace self-blame when LMX quality was high (ρ = 0.17, SE = 0.02, 95%CI [0.13 TO 0.20]) than when LMX quality was low (ρ = 0.11, SE = 0.02, 95%CI [0.08 TO 0.16]). Their difference was significant (Δρ = 0.05, SE = 0.02, 95%CI [0.01 TO 0.10]). Thus, Hypothesis 2 was supported. 

Hypothesis 3 predicted that LMX moderated the indirect effect of abusive supervision on supervisor-directed helping via the chain mediators of workplace self-blame and workplace guilt. The indirect effect was more negative when LMX quality was high than when LMX quality was low. Table 4 showed that abusive supervision had a more negative indirect effect on supervisor-directed helping via workplace self-blame and workplace guilt when LMX quality was high (ρ = −0.011, SE = 0.005, 95%CI [−0.021 TO −0.003]) than when LMX quality was low (ρ = −0.008, SE = 0.003, 95%CI [−0.016 TO −0.002]). Their difference was significant (ρ = −0.003, SE = 0.002, 90%CI [−0.008 TO −0.001]). Thus, Hypothesis 3 was supported.

### 4.4. Test of Multilevel Moderated Mediation Model 

Hypothesis 4 predicted that SLMX moderated the relationship between workplace guilt and supervisor-directed helping such that their relationship was more negative when SLMX quality was low (vs. high). Table 5 (see Model 3) showed a significant interaction effect between workplace guilt and SLMX on supervisor-directed helping (*γ* = 0.12, SE = 0.05, *p* < 0.05). Table 6 (upper part) showed that workplace guilt was more negatively related to supervisor-directed helping when SLMX was low-quality (*γ* = −0.12, SE = 0.05, *p* < 0.05) than when SLMX quality was high (*γ* = 0.03, SE = 0.04, ns.). Their difference was significant (*γ* = −0.14, SE = 0.06, *p* < 0.05). Thus, Hypothesis 4 was supported.

We displayed the conditional effect of workplace guilt on supervisor-directed helping at the full range of SLMX from low-quality to high-quality using the Johnson–Neyman technique. As shown in Figure 3, the vertical line (the value of SLMX = −0.30) indicated the boundary between the significant and non-significant regions for the conditional effect of workplace guilt on supervisor-directed helping. When the value of SLMX was below −0.30, the effect of workplace guilt on supervisor-directed helping was significant. In addition, Figure 3 also showed that when the value of SLMX was above 2.0, workplace guilt was positively related to supervisor-directed helping.

Hypothesis 5 predicted that the relationship between abusive supervision and supervisor-directed helping is serially mediated by workplace self-blame and workplace guilt. The indirect effect was more negative when LMX quality was high (vs. low) but SLMX quality was low (vs. high). Table 6 (lower part) showed that abusive supervision had more negative indirect effect on supervisor-directed helping via workplace self-blame and workplace guilt for pattern (1) (LMX quality was high and SLMX quality was low) (ρ = −0.025, SE = 0.011, 95% Monte Carlo CI [−0.031 TO −0.002]) than for other patterns. These include pattern 2 (both LMX quality and SLMX quality were low), pattern 3 (both LMX quality and SLMX quality were high), and pattern 4 (LMX quality was low, but SLMX quality was high). The differences of pattern 1 from pattern 2, pattern 3, and pattern 4 were significant, respectively (Δρ = −0.025, SE = 0.012, *p* = 0.036; Δρ = −0.031, SE = 0.041, *p* = 0.027; Δρ = −0.025, SE = 0.011, *p* = 0.024). Thus, Hypothesis 5 was supported. By adding the effects of LMX and its interaction with abusive supervision onto workplace guilt, the complete results remain unchanged.

## 5. Discussion

Our study aims to reevaluate previous speculations on the beneficial consequences of abusive supervision for supervisors. We replicate, differ from, and extend Tröster and Van Quaquebeke’s (2021) research on the impact of abusive supervision on supervisor-directed helping [9]. First, we replicated two of their findings based on the subordinate perspective. Consistently, LMX strengthens the positive effect of abusive supervision on workplace self-blame, and the positive indirect effect of abusive supervision on workplace guilt through workplace self-blame. Second, we have a different finding from theirs. When supervisor-directed helping is rated by supervisors themselves, abusive supervision has a negative indirect effect on supervisor-directed helping (serially mediated by workplace self-blame and workplace guilt), and the indirect effect is stronger when LMX quality is higher. Third, we extended their study and found that SLMX moderates the relationship between workplace guilt and supervisor-directed helping such that their negative relationship is stronger when SLMX quality is low (vs. high). Finally, we found that the indirect effect of abusive supervision on supervisor-directed helping is more negative when LMX quality is higher, but SLMX quality is lower. 

Theoretically, our study contributes to the literature on abusive supervision in reevaluating seemingly positive or beneficial effect of abusive supervision. Our study extends Tröster and Van Quaquebeke’s (2021) work by integrating both subordinates’ and supervisors’ perspectives. Taking the supervisor perspective allows us to reevaluate previous speculations on the beneficial consequences of abusive supervision for supervisors. As warned by [12], “inducing guilt appears to be a potentially costly technique for getting one’s way and overdoing it can be extremely destructive” (p. 263). By contrast, our findings are consistent with prior meta-analysis and review studies that subordinate-reported abusive supervision has negative direct and indirect effect on the supervisor’s ratings of their subordinates’ in-role and extra-role performance. This study, together with Pan et al. (2018) [8], suggests that abusive supervision creates a negative employee attitude and has unbeneficial consequences for leaders. Second, the study provides compelling evidence that the validity of academic research can be enhanced through more rigorous empirical tests [29]. This is important because “creating theory without ever testing it stringently or testing it only once and accepting it as gospel is also unhelpful” [29] (p. 510). 

Practically, our study has an important implication for managers. Our study warns those supervisors who assume that workplace guilt may act as an interpersonal influence technique to dominate their subordinates. Although mutual concern and respect are the basis of close relationships, subordinates tend to be more willing to contribute to the relationship due to the fact that they have less power. That is why, in reality, they may feel guilty even when they did nothing wrong. Thus, taking their advantage in the relationship, supervisors can create in the subordinates an affective state of guilt to motivate them to do what they desire. It has been found that workplace guilt may operate as an influence technique that lets relatively powerless subordinates make more contributions [12]. However, our study indicates that such workplace guilt is essentially out of mutual concern and for promoting interpersonal equity. While the abusive behavior increasingly weakens the quality of LMX, workplace guilt will lose its root, and the assumed link found in Tröster and Van Quaquebeke’s (2021) study will disappear. As suggested by [30], if it is leaders who deliberately use abusive supervision to satisfy their personal interests, their behaviors can backfire and harm their reputation and credibility and erode their social capital.

For future research, scholars should avoid treating abusive supervision as acceptable to subordinates or beneficial to supervisors in some contexts. Future research may investigate how abusive supervision triggers subordinates’ positive and constructive reactions. Could abused subordinates enhance their performance because of their dissatisfaction with such treatment or because of their need for self-esteem and dignity? Tepper et al. (2017) proposed that abusive supervision can be performance-enhancing and influence employee work outcomes through performance-enhancing pathways, such as attention, desire to avoid further hostility, and desire to prove the supervisor wrong [4].

## 6. Conclusions

Do victims really help their abusive supervisors? Does abusive supervision yield any positive consequence? Using subordinates’ self-reports, Tröster and Van Quaquebeke (2021) found that abusive supervision in high-quality LMX relationships lets subordinates blame themselves, subsequently feel guilty, and make up for it by being more helpful. However, our study suggests that supervisors evaluate their abused subordinates as less helpful. The contrasting effect of abusive supervision becomes more striking when subordinates perceive a high-quality LMX relationship, while supervisors maintain a low-quality SLMX relationship with them. In contrast to previous speculations, the abused subordinates make positive and constructive reactions to enhance their performance, which can be conceived of as improving the relationship quality with their supervisors or to retaining their dignity.

## 7. Supplementary Data Analysis (1)

To cross-validate, to some extent, our major research findings, we conducted additional data analysis on whether the negative indirect effect of abusive supervision on supervisor-directed helping for low-quality SLMX could still be established when abusive supervision was reported by supervisors themselves. At Time 1 of the study, we asked supervisors to report the frequency at which they tend to use abusive behaviors towards their subordinates. We used the five-item scale, originally developed by [5], and later adapted by [18]. We used “I” to replace “my supervisor” in the original version. One sample item was “I might ridicule my subordinates”. The five-point Likert scale ranged from 1 (never) to 5 (always). Alpha reliability of the scale was 0.89.

In data analysis, supervisor-reported abusive supervision and SLMX were grand-mean centered, and these two variables and their interaction term were specified at group level. The interaction effect of abusive supervision and SLMX on supervisor-directed helping was serially mediated by workplace self-blame and workplace guilt, and the effect of workplace guilt on supervisor-directed helping was further moderated by SLMX. Results showed that the indirect effect of supervisor-reported abusive supervision on supervisor-directed helping via the two chain mediators was more negative when SLMX quality was low (ρ = −0.038, SE = 0.022, 90% Monte Carlo CI [−0.074 TO −0.002]) than when SLMX quality was high (ρ = 0.000, SE = 0.003, 90% Monte Carlo CI [−0.006 TO 0.006]. Their difference was significant (90% Monte Carlo CI [−0.074 TO −0.002]). In conclusion, abusive supervisors did not anticipate their subordinates to engage in voluntary behavior that benefits them. Indeed, they rated their subordinates as less helpful, particularly when they maintained low-quality relationships with their subordinates.

## 8. Supplementary Data Analysis (2)

Additionally, we tested our cross-level moderated mediation model using grand-mean centering exogenous variables in the model, suggested by [31]. For hypothesis 4, concerning the moderating effect of SLMX on the relationship between workplace guilt and supervisor-directed helping, the results were the same as those of prior model testing.

The interaction term between abusive supervision and LMX on workplace self-blame was 0.12 (SE = 0.07, *p* = 0.092). The effect of workplace self-blame on workplace guilt was 0.30 (SE = 0.04, *p* = 0.000). For Hypothesis 5, abusive supervision had a more negative indirect effect on supervisor-directed helping via workplace self-blame and workplace guilt for pattern (1) (LMX quality was high and SLMX quality was low (ρ = −0.020, SE = 0.009, 95% Monte Carlo CI [−0.039 TO −0.003]) than for other patterns. These include pattern 2 (both LMX quality and SLMX quality were low) (ρ = −0.014, SE = 0.007, 95% Monte Carlo CI [−0.029 TO −0.002]), pattern 3 (both LMX quality and SLMX quality were high) (ρ = 0.004, SE = 0.007, 95% Monte Carlo CI [−0.009 TO 0.018]), and pattern 4 (LMX quality was low but SLMX quality was high) (ρ = 0.003, SE = 0.005, 95% Monte Carlo CI [−0.007 TO 0.012]). The differences of pattern 1 from pattern 3 and pattern 4 were significant (Δρ = −0.024, SE = 0.011, *p* = 0.027; Δρ = −0.023, SE = 0.010, *p* = 0.019), but the difference between pattern 1 and pattern 2 was not significant (Δρ = −0.006, SE = 0.005, *p* = 0.181). By adding the effects of LMX and its interaction with abusive supervision onto workplace guilt, the complete results remain unchanged.

## Figures and Tables

**Figure 1 behavsci-13-00815-f001:**
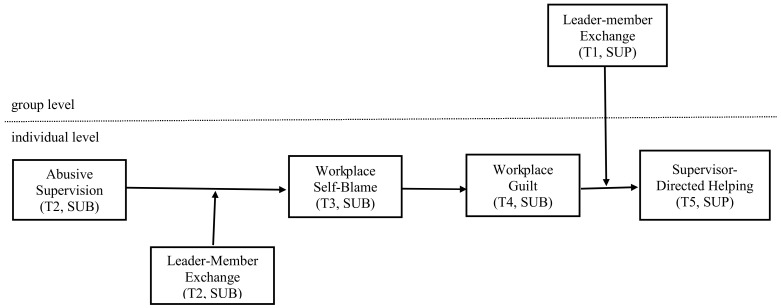
Research model of the study. Note: T = time; SUP = supervisor, SUB = subordinate.

**Figure 2 behavsci-13-00815-f002:**
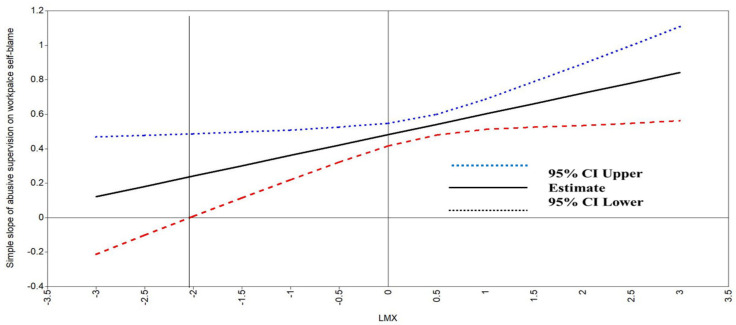
Region of significance for the simple slope of abusive supervision on workplace guilt.

**Figure 3 behavsci-13-00815-f003:**
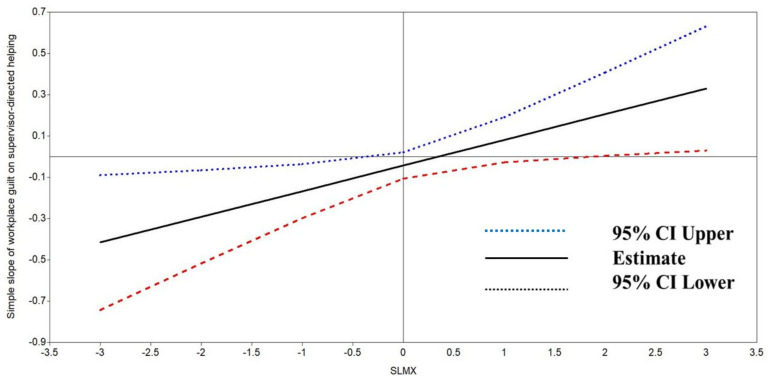
Region of significance for the simple slope of workplace guilt on supervisor-directed helping.

**Table 1 behavsci-13-00815-t001:** Descriptive statistics and correlations.

Variables	Mean	SD	1	2	3	4	5	6
Individual level								
**1. SLMX** (T1, Sup−reported)	3.93	0.57	(0.83)					
2. Abusive supervision (T2, Sub−reported)	2.13	1.12	−0.02	(0.94)				
3. LMX (T2, Sub−reported)	3.74	0.73	0.32 **	0.09 **	(0.86)			
4. Workplace self−blame (T3, Sub−reported)	2.75	1.23	−0.01	0.48 **	0.10 **	(0.90)		
5. Workplace guilt (T4, Sub−reported)	2.95	1.02	−0.04	0.47 **	0.09 **	0.50 **	(0.90)	
6. Sup−directed helping (T5, Sup−rated)	3.91	0.74	0.24 **	−0.00	0.20 **	−0.03	−0.06 *	(0.85)
**Group level**								
SLMX	3.90	0.58	(0.84)					

*n* = 1002 (individual level), 189 (group level); reliabilities are on the diagonal. * *p* < 0.05, ** *p* < 0.01 Sup = supervisor, Sub = subordinate.

**Table 2 behavsci-13-00815-t002:** Comparison of measurement models.

Models	x^2^	df	∆x^2^	RMSEA	CFI	TLI	SRMR
6-factor (Baseline) Model	1321.52	335		0.054	0.94	0.93	0.046
5-factor Model 1 (SLMX + LMX)	2388.76	340	1067.24 **	0.078	0.88	0.86	0.066
5-factor Model 2 (Abusive supervision + LMX)	3831.86	340	2510.34 **	0.101	0.79	0.77	0.121
5-factor Model 3 (Abusive supervision + Workplace blame)	2866.10	340	1544.58 **	0.086	0.85	0.83	0.070
5-factor Model 4 (Workplace blame + Workplace guilt)	2681.47	340	1359.95 **	0.083	0.86	0.84	0.062
5-factor Model 5 (SLMX + Sup-directed helping)	2514.02	340	1192.5 **	0.080	0.87	0.86	0.073
5-factor Model 6 (Workplace guilt + Sup-directed helping)	3269.38	340	1947.86 **	0.093	0.82	0.80	0.102

** *p* < 0.01.

**Table 3 behavsci-13-00815-t003:** Unstandardized coefficients and standard errors of single-level moderated mediation path analysis.

	Workplace Self-Blame		Workplace Guilt		Supervisor-Directed Helping	
	Model 1		Model 2		Model 3	
	*b*	SE	*b*	SE	*b*	SE
Workplace guilt					−0.07 **	
Workplace self-blame			0.29 ***		−0.02	0.03
Abusive supervision	0.48 ***		0.23 ***	0.03	−0.004	0.02
LMX	0.10	0.03	0.06	0.03	0.22 ***	0.03
Abusive supervision × LMX	0.12 *	0.06	0.14 **	0.04	0.09 *	0.04
R^2^	0.24	0.05	0.33	0.04	0.05	0.04

*n* = 1002, * *p* < 0.05, ** *p* < 0.01, *** *p* < 0.001.

**Table 4 behavsci-13-00815-t004:** Results of single-level moderated indirect effect of abusive supervision.

Dependent Variable	Mediator	Moderator	Effect	SE	
**Moderating effect**					*p*
DV = workplace self-blame		High-quality LMX	b = 0.57	0.04	0.000
		Low-quality LMX	b = 0.39	0.06	0.000
		Difference	Δb = 0.18	0.07	0.018
**Moderated indirect effect** **(with single mediator)**					95% bias−corrected CI
DV = workplace guilt	workplace self-blame	High-quality LMX	ρ = 0.17	0.02	0.13 TO 0.20
		Low-quality LMX	ρ = 0.11	0.02	0.08 TO 0.16
		Difference	Δρ = 0.05	0.02	0.01 TO 0.10
**Moderated indirect effect** **(with chain mediators)**					95% bias−corrected CI
DV = supervisor-directed helping	workplace self-blame and workplace guilt in sequence	High-quality LMX	ρ = −0.011	0.005	−0.021 TO −0.003
		Low-quality LMX	ρ = −0.008	0.003	−0.016 TO −0.002
		Difference	Δρ = −0.003	0.002	−0.008 TO −0.001 (90%CI)

*n* = 1002.

**Table 5 behavsci-13-00815-t005:** Results of multilevel path analysis: the estimated direct effects and interaction effects.

	Workplace Self-Blame		Workplace Guilt		Supervisor-Directed Helping	
	Model 1		Model 2		Model 3	
	*γ*	SE	*γ*	SE	*γ*	SE
**Level 2 Independent Variables** Workplace guilt SLMX Workplace guilt × SLMX					−0.04 −0.06 0.12 *	0.03 0.17 0.05
**Level 1 Independent Variables** Abusive supervision LMX Abusive supervision × LMX Workplace self-blame	0.26 *** 0.14 0.56 ***	0.06 0.09 0.10	0.14 ** 0.41 ***	0.05 0.04	0.03 0.00	0.04 0.03
Level-2 residual variance (*τ*) Level-1 residual variance (*σ*^2^)	10.44		0.77		0.28 0.38	

*n* = 1002 (individual level), *n* = 189 (group level). * *p*< 0.05, ** *p* < 0.01, *** *p* < 0.001.

**Table 6 behavsci-13-00815-t006:** Results of cross-level moderating and moderated indirect effects.

Independent Variable	Mediator	Individual-Level Moderator	Group-Level Moderator	Effect	SE	
**Moderating effect**				*γ*		*p*
DV = supervisor-directed helping			High-quality SLMX	0.03	0.04	0.464
			Low-quality SLMX	−0.12 *	0.05	0.019
			Difference	−0.14 *	0.06	0.017
**Moderated indirect effect**				ρ		95% Monte Carlo CI
DV = supervisor-directed helping	workplace self-blame and workplace guilt in sequence	P1: High-quality LMX	Low-quality SLMX	−0.025	0.011	[−0.031 TO −0.002]
		P2: Low-quality LMX	Low-quality SLMX	0.000	0.004	[−0.008 TO 0.009]
		P3: High-quality LMX	High-quality SLMX	0.006	0.009	[−0.006 TO 0.016]
		P4: Low-quality LMX	High-quality SLMX	0.000	0.001	[−0.003 TO 0.004]
**Difference of indirect effect**				Δρ	SE	*p*
		P1 − P2		−0.025	0.012	0.036
		P1 − P3		−0.031	0.041	0.027
		P1 − P4		−0.025	0.011	0.024

* *p* < 0.05. P = pattern.

## Data Availability

Data Availability Statement: Datasets analyzed or generated during the study are under the control of the investigators at the correspondence author.

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
