# Peer review of "Do Victims Really Help Their Abusive Supervisors? Reevaluating the Positive Consequences of Abusive Supervision"

_behavsci, 2023, doi:10.3390/bs13100815_

Round 1
Reviewer 1 Report
The article is well written and easy to read and understand. However, it is recommended that before publication, the authors better relate the theory that supports each hypothesis.
That is, theoretical arguments 1, then hypothesis 1. Theoretical arguments 2, then hypothesis 2.
Author Response
Dear reviewer 1,
Please read our response to comments by you and all other reviewers in the file attached.
Authors

Reviewer 2 Report
Thanks for allowing me to review this manuscript. Due to the interesting topic and professional writing, I enjoyed reading the manuscript. I have no particular issues and I find the manuscript at a very good stage. With potential readers in mind, I would suggest reviewing only one paragraph within the manuscript. The paragraph from lines 110 to 143 appears to be quite long and I suggest the authors reduce the number of words or split the paragraph into two sections by focusing on the different hypotheses. I think the rest is very well written and I wish to compliment the authors for their work.
Very good English. No comments
Author Response
Dear reviewer 2,
Please read our response to comments by you and other reviewers in the attached file.
Authors

Reviewer 3 Report
General comments:
As always I started my review with a general analysis of this research article in the context of elements such as: a structure of the article, abstract, introduction, literature review, and methodology, the results of the research, discussion, conclusions, limitations and future directions of the research. I found these elements in the article but not all (Authors should prepare also Conclusions at the end of the article and this’s necessary to describe also Limitations and Future research directions).
In my opinion, the Authors undertake actual and interesting problem related to the topic - If Victims Really Help Their Abusive Supervisors? The Authors analyzed this topic from the perspective of Supervisors’. The content of the article is coherent with the topic. Well-formulated hypotheses.
The cited references are relevant to the topic of the article but the Bibliography must be longer, more sources are needed (25 sources is not enough in this kind of article in my opinion). The Authors should show a greater knowledge of sources related to the analyzed issues (min. 50 sources).
The manuscript’s results are presented clearly and understandably but below all tables the Authors should write - the source (e.g., Source: Own elaboration).There are no Conclusions in this article (as a separate Section); the Conclusions must be consistent with the evidences and arguments presented in the article. The Authors should also describe the future directions of research related to the analyzed issues.
Notes to be included in fixes:
-In Abstract it should be always written the main aim/purpose of the article.
-In the introduction (in the research article), the Authors should clearly state the purpose of the article and also the research question(s)/research problem(s).
- The results of the conducted literature research on the analyzed issues are clearly presented with reference to different publications but in my opinion, the Authors should use much more sources and the bibliography could be longer; min. 50 sources, especially the Authors must use the latest articles (2018-2023 in the analyzed area).
- Below all tables the Authors should write – the source (e.g., Source: Own elaboration).
-After the section Discussion must be also prepare the Section – Conclusion(s) and the Authors should also write the Section about Limitations and Future Research directions (it’s necessary).
Minor editing of English language required.
Author Response
Dear reviewer 3,
Please read our response to comments by you and other reviewers in the attached file.
Authors
